# Systemic Adverse Events and Use of Antipyretics Predict the Neutralizing Antibody Positivity Early after the First Dose of ChAdOx1 Coronavirus Disease Vaccine

**DOI:** 10.3390/jcm10132844

**Published:** 2021-06-27

**Authors:** Ji Young Park, Seong-Ho Choi, Jin-Won Chung, Min-Hyung Hwang, Min-Chul Kim

**Affiliations:** 1Department of Pediatrics, Chung-Ang University Hospital, Seoul 06973, Korea; jypark@caumc.or.kr; 2Department of Internal Medicine, Division of Infectious Diseases, Chung-Ang University Hospital, Seoul 06973, Korea; tobeservant@cau.ac.kr (S.-H.C.); hhmin6975@gmail.com (M.-H.H.); kimminchulmd@caumc.or.kr (M.-C.K.)

**Keywords:** neutralizing antibodies, health care workers, SARS-CoV-2, vaccination

## Abstract

Vaccination is considered crucial for the eradication of the coronavirus disease (COVID-19). In our medical center in Korea, most health care workers (HCWs) were vaccinated with the ChAdOx1 COVID-19 vaccine. After vaccination, many HCWs complained of adverse events (AEs). However, it remains unclear whether the production of neutralizing antibodies (NAb) was affected. Therefore, here, we aimed to evaluate AEs and early NAb production in relatively healthy Asians who received the ChAdOx1 vaccine and determine the effect of AEs and antipyretics on early NAb production against COVID-19. Of the 182 Korean HCWs who received the first dose of ChAdOx1 vaccine, 172 (94.5%) experienced ≥1 adverse events and 148 (81.3%) tested positive for NAb 33–40 days after the vaccination. NAb-positive vaccine recipients reported systemic AEs and consumed acetaminophen more frequently than NAb-negative recipients. We identified an association between antibody response and COVID-19 vaccine-related AEs. In conclusion, most ChAdOx1 vaccine recipients reported AEs in our medical center.

## 1. Introduction

The coronavirus disease (COVID-19) is an ongoing global pandemic caused by severe acute respiratory syndrome coronavirus 2 (SARS-CoV-2). Countries worldwide have emphasized the need for social distancing, wearing a facemask, and proper personal hygiene to prevent the spread of SARS-CoV-2 in the community. While the role of non-pharmaceutical interventions in preventing disease transmission has limitations, vaccination is crucial for the eradication of COVID-19. In Korea, vaccination began on 26 February 2021, starting with health care workers (HCWs) and patients in nursing hospitals. In our medical center, most HCWs were vaccinated with the ChAdOx1 COVID-19 vaccine. After vaccination, many HCWs complained of adverse events (AEs) with varying characteristics and severity [1,2,3]. However, there was little reference data on the AEs because Asians were only <5% of the total participants enrolled in the clinical trials for the ChAdOx1 SARS-CoV-2 vaccine [4,5]. In a previous study, inflammation-related AEs indicated a stronger immune response following vaccination [6]. Despite the frequent occurrence of AEs and common use of symptomatic drugs such as antipyretics, it remains unclear whether there were any changes in the production of neutralizing antibodies (NAb).

Therefore, the present study aimed to evaluate AEs and early NAb production in relatively young and healthy Asians who received the ChAdOx1 vaccine and determine the effect of AEs and antipyretics on early NAb production against COVID-19.

## 2. Materials and Methods

### 2.1. Study Design and Data Collection

This study was conducted as a cross-sectional study. Among the 1356 HCWs who received the first dose of the ChAdOx1 vaccine (AstraZeneca/Oxford) at the Chung-Ang University Hospital, we enrolled HCWs who participated voluntarily in this study and excluded those who received the BNT162b2 vaccine (Pfizer/BioNTech, USA) as well as those diagnosed with COVID-19. We collected blood samples from participants 33–40 days after the first dose of vaccine. Demographics and AE data were collected through questionnaires. The acquired data included the following: sex, date of birth/vaccination, occupation, history of COVID-19/drug AEs/allergy, severity/duration of AEs, and the duration/use of acetaminophen. Severity was subjectively graded from 1 to 5 according to the Faces Pain Scale. Each AE was classified into systemic and localized. The severity of symptoms was scored in two ways: the sum of each symptoms’ severity score (SUM) and the sum of multiplying each symptoms’ severity by the duration (days) (SoM = ∑ (symptom’s severity × days)).

### 2.2. Serological Assays

Circulating NAbs were detected using the GenScript SARS-CoV-2 surrogate virus neutralization test (sVNT) kit (Genscript Biotech Corporation, Piscataway, NJ, USA). To enhance the reliability of the experiments, the study samples were also tested using the Euroimmun anti-SARS-CoV-2 IgG enzyme-linked immunosorbent assay (ELISA) (Euroimmun, Lübeck, Germany). Both assays were performed according to the manufacturer’s guidelines. The IgG results were either positive (index ≥ 1.1), borderline, or negative (index < 0.8). The borderline IgG ELISA result was categorized as negative in this study. The sVNT kits’ results were interpreted by the inhibition rate, which was calculated as follows:$$\mathrm{Inhibition}=\left(1-\frac{\mathrm{Optical}\;\mathrm{density}\;\mathrm{value}\;\mathrm{of}\;\mathrm{sample}}{\mathrm{Optical}\;\mathrm{density}\;\mathrm{of}\;\mathrm{negative}\;\mathrm{control}}\right)\;\times \;100$$

It was classified into positive and negative samples with a 30% cutoff [7].

### 2.3. Statistical Analysis

All statistical analyses were performed using IBM SPSS Statistics for Windows, version 25.0 (IBM Corp., Amonk, NY, USA). Categorical variables were analyzed using the χ^2^ test, Student’s *t*-test, one-way ANOVA, and Fisher–Freeman–Halton test. The Pearson correlation coefficients were computed for the normally distributed data, and the Spearman’s rank correlation coefficients were computed for the non-normally distributed data. A regression model was also considered. A *p*-value < 0.05 indicated statistical significance.

### 2.4. Ethics Statement

The institutional review board (IRB) of Chung-Ang University Hospital approved this study (IRB No. 2051-001-415). Written consent was obtained from all enrolled participants.

## 3. Results

Between 4 March 2021 and 10 March 2021, 1356 HCWs were vaccinated with the first dose of ChAdOx1 vaccine at our hospital. Of those, 182 (13.4%) participated in this study. None of the participants reported serious AEs. All blood samples were analyzed using sVNT, and 180 were analyzed using Euroimmun ELISA.

### 3.1. Demographics and Vaccine Adverse Events

Of the 182 participants, 117 (64.3%) were female, and the mean age was 38.0 (range: 23–63) years; 85 (46.7%) were doctors, 69 (37.9%) were nurses, and 22 (12.1%) were medical laboratory technologists. Researchers, radiologists, and hospital administrative assistants accounted for 1.1% (*n* = 2 each) of the participants. No participant had a history of COVID-19. Ten (5.5%) participants had a history of drug-related AEs and 15 (8.2%) had a history of allergy. Nearly all (172, 94.5%) of the participants reported at least one AE. The most common AE was muscle or joint pain (73.6%), followed by injection site pain (69.2%), fatigue (67.0%), chills (64.3%), fever (51.1%), and headache (42.9%). The use of acetaminophen was observed in 81.9% of the participants (Table 1).

### 3.2. Association between Demographics and Scores of Adverse Events

The systemic and localized SUM/SoM showed significant differences by sex (*p* = 0.001, < 0.001, 0.004, and 0.006, respectively) but not by occupation (*p* = 0.159, 0.763, and 0.626, respectively), except for a difference in the systemic SoM between nurses and medical laboratory technologists (*p* = 0.028). The association between demographics and AEs is presented in Table 2. The systemic SUM and SoM showed a moderate negative linear relationship by aging (coefficient = −0.356 and −0.305, *p* < 0.001 for all), whereas the localized SUM and SoM showed a weak negative linear relationship by aging (coefficient = −0.169 and −0.160, *p* = 0.023 and 0.031, respectively).

### 3.3. NAb Positivity and Related Factors

Of the total 182 samples, positive NAb was found in 148 (81.3%). Both the positive and negative results of NAb did not differ significantly with sex, age, and occupation (*p* = 0.054, 0.784, and 0.124, respectively). Of the AE symptoms, participants with the following signs showed significantly more positive than negative NAb results: fever (55.4% vs. 32.4%; *p* = 0.015), headache (46.6% vs. 26.5%; *p* = 0.032), chills (68.9% vs. 44.1%; *p* = 0.006), muscle/joint pain (77.7% vs. 55.9%; *p* = 0.009), and fatigue (72.3% vs. 44.1%; *p* = 0.002). However, NAb levels in patients with fever and headache were not different when we performed IgG ELISA (*p* = 0.112 and 0.160). A significantly greater proportion of vaccine recipients in the NAb-positive group consumed acetaminophen than those in the NAb-negative group (*p* = 0.017). The positive rate with sVNT significantly increased when systemic SUM and SoM were higher (*p* = 0.004 and 0.010) but showed no difference in the localized SUM/SoM (*p* = 0.199 and 0.122) for IgG ELISA (Table 1).

NAb levels showed significant differences based on sex (*p* = 0.038) but no significant difference based on occupation and age (*p* = 0.761 and 0.227). The values of NAb increased significantly with higher systemic SUM/SoM (*p* = 0.003 and 0.006; Figure 1). However, no difference was found for the localized SUM/SoM (*p* = 0.301 and 0.369). The IgG level showed a difference based on sex (*p* = 0.009) but not based on occupation and age (*p* = 0.508 and 0.230). IgG levels also increased with a higher systemic SUM/SoM (*p* = 0.002 and 0.009; Figure 1), but no difference was observed in terms of localized SUM/SoM (*p* = 0.194 and 0.162).

### 3.4. Antibody Values Using Two Commercial Kits

Of the 180 samples, positive results were reported in 146 (81.1%) participants who were tested using sVNT kits and in 162 (90.0%) tested using IgG ELISA kits. When IgG ELISA results that were borderline or negative were categorized as negative results, the strength of agreement between the two assays was the highest (κ = 0.602, *p* < 0.001; Table 3). The values between IgG ELISA and sVNT showed a strongly positive linear correlation (r = 0.882, R2 = 77.7%, *p* < 0.001; Figure 2).

## 4. Discussion

In Korea, AE-reporting rates were approximately 90%, more than those reported in phase 1/2 clinical trials [1,2,3,5]. According to the Yellow Card reporting, which is a voluntary AE-reporting scheme in the United Kingdom, the overall AE-reporting rate is around 3–6 Yellow Cards per 1000 doses administered [8]. A previous study reported a higher incidence of fever and flu-like illness after vaccination in a non-white ethnicity population [9].

Notwithstanding the frequent use of antipyretics due to common AEs, the early antibody response in our study was 81.3%, which was better than the efficacy of the phase 3 clinical trial (64.1%) after the first standard dose [4]. Only a small number of Asians participated in previous clinical trials; therefore, limited information was available regarding AEs and immune response in Asians [4,5]. Because of many AEs, most vaccine recipients consumed antipyretics, which could be associated with blunt immune response. According to a literature review, observational studies showed that antipyretic use did not affect antibody response; however, a few randomized clinical trials have reported a dampened immune response of unknown clinical significance [10]. At present, antibody response remains unclear because no study has so far reported on the effect of antipyretics after COVID-19 vaccination.

We also found that systemic and localized AE scores (SUM and SoM) were higher among females than among males. This phenomenon is attributed to unknown immunological differences between the two sexes [1,11].

This study has several limitations. First, among HCWs enrolled in this study, some may have had COVID-19 that remained undetected during symptom-based testing, which may have affected antibody test results. However, in Korea, only 0.04% and 0.18% were diagnosed with COVID-19 even in large-scale sero-surveillance, perhaps due to the small scale of the COVID-19 outbreak in Korea due to thorough infection control and prevention. Moreover, from March 21, all HCWs in Seoul were instructed to undergo PCR tests every two weeks. Therefore, it is considered that the probability of COVID-19 breakthrough infection among HCWs is very low. Second, the severity of AEs in each individual was reported subjectively and the participants volunteered to be included in the study, which might have led to bias. Third, the sample size was small and the study was conducted with HCWs at a single medical center; thus, the results can hardly be attributed to the general Korean population. Finally, because higher antibody values do not imply better protection from SARS-CoV-2, the results must be interpreted cautiously. Despite these limitations, the present study has several strengths. We identified an association between antibody response and COVID-19 vaccine-related AEs. Moreover, this is the first study to show no association between the antibody response and acetaminophen use to mitigate COVID-19 vaccine-related symptoms.

## 5. Conclusions

Most ChAdOx1 vaccine recipients reported AEs in this study. Satisfactory early antibody response was observed, especially among vaccine recipients with more or severe AEs. There was no evidence supporting blunted antibody response with the use of acetaminophen. Therefore, acetaminophen can be considered for active symptomatic management of AEs.

## Figures and Tables

**Figure 1 jcm-10-02844-f001:**
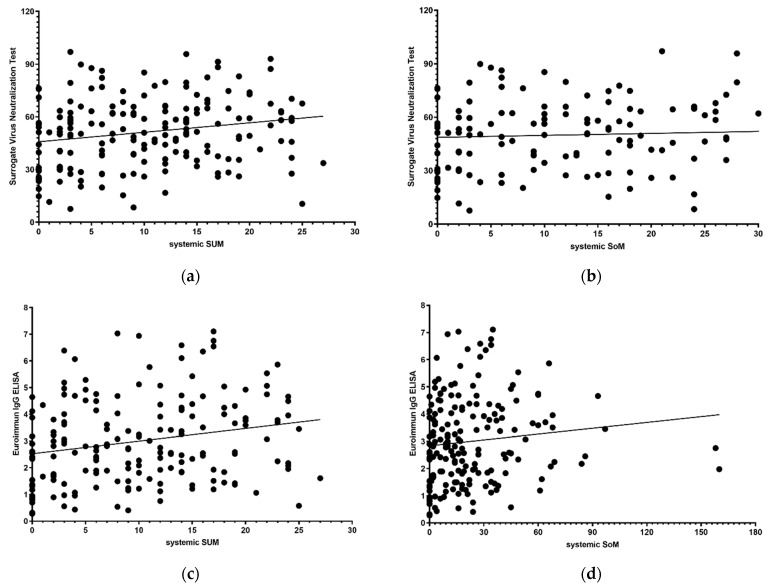
Antibody response by scores of systemic adverse events. (**a**) Values of surrogate virus neutralization test by systemic SUM; (**b**) values of surrogate virus neutralization test by systemic SoM; (**c**) values of Euroimmun IgG ELISA by systemic SUM; (**d**) values of Euroimmun IgG ELISA by systemic SoM. Abbreviations: ELISA, enzyme-linked immunosorbent assay; SUM, sum of symptoms’ severity score; SoM, sum of multiplying each symptoms’ severity by the duration of symptoms.

**Figure 2 jcm-10-02844-f002:**
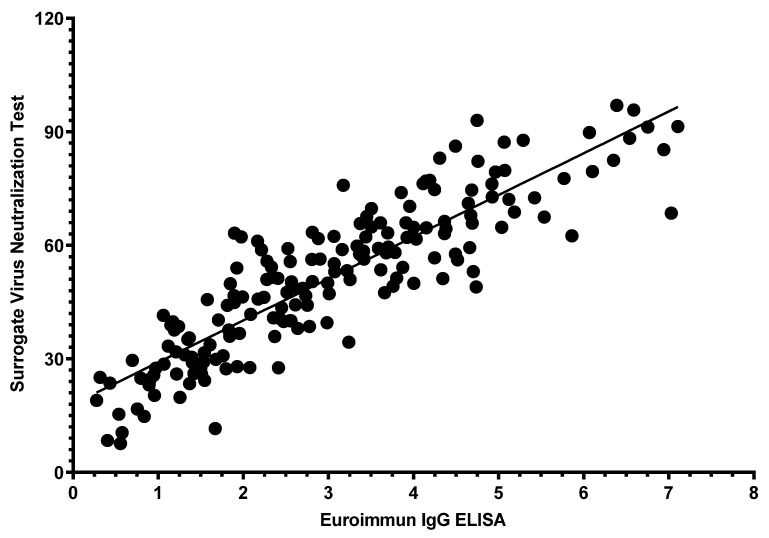
Correlation of values between Euroimmun IgG ELISA and surrogate virus neutralization test. Abbreviations: ELISA, enzyme-linked immunosorbent assay.

**Table 1 jcm-10-02844-t001:** Demographics and vaccine adverse event serological assays.

Variables	Surrogate Virus Neutralization Test			EUROIMMUN IgG ELISA			Total
	Positive, *n* = 148	Negative, *n* = 34	*p*	Positive, *n* = 162	Negative, *n* = 18	*p*	
Sex			0.054			0.376	
Male	48 (32.4)	17 (50.0)		55 (34.0)	8 (44.4)		65 (35.7)
Female	100 (67.6)	17 (50.0)		107 (66.0)	10 (55.6)		117 (64.3)
Ages, years (mean ± SD)	37.9 ± 10.7	38.4 ± 10.0	0.784	37.4 ± 10.3	41.3 ± 10.1	0.126	38.0 ± 10.5
Occupation			0.124			0.137	
Doctor	71 (48.0)	14 (41.2)		77 (47.5)	6 (33.3)		85 (46.7)
Nurse	59 (39.9)	10 (29.4)		62 (38.3)	7 (38.9)		69 (37.9)
Laboratory technologist	14 (9.5)	8 (23.5)		19 (11.7)	3 (16.7)		22 (12.1)
Researcher	1 (0.7)	1 (2.9)		1 (0.6)	1 (5.6)		2 (1.1)
Radiologist	2 (1.4)	0		2 (1.2)	0		2 (1.1)
Hospital administrative assistant	1 (0.7)	1 (2.9)		1 (0.6)	1 (5.6)		2 (1.1)
History of COVID-19	0	0		0	0		0
History of drug adverse event	8 (5.4)	2 (5.9)	>0.999	9 (5.6)	1 (5.6)	>0.999	10 (5.5)
History of allergy	12 (8.1)	3 (8.8)	>0.999	13 (8.0)	2 (11.1)	0.649	15 (8.2)
Vaccine adverse events							
Systemic							
Fever	82 (55.4)	11 (32.4)	0.015 *	86 (53.1)	6 (33.3)	0.112	93 (51.1)
Headache	69 (46.6)	9 (26.5)	0.032 *	73 (45.1)	5 (27.8)	0.160	78 (42.9)
Chills	102 (68.9)	15 (44.1)	0.006 *	111 (68.5)	5 (27.8)	0.001 *	117 (64.3)
Nausea	24 (16.2)	2 (5.9)	0.174	25 (15.4)	1 (5.6)	0.478	26 (14.3)
Vomiting	4 (2.7)	0	>0.999	4 (2.5)	0	>0.999	4 (2.2)
Diarrhea	10 (6.8)	0	0.212	10 (6.2)	0	0.601	10 (5.5)
Muscle/joint pain	115 (77.7)	19 (55.9)	0.009 *	126 (77.8)	7 (38.9)	<0.001 *	134 (73.6)
Fatigue	107 (72.3)	15 (44.1)	0.002 *	115 (71.0)	6 (33.3)	0.001 *	122 (67.0)
Localized							
Pain at injection site	102 (68.9)	24 (70.6)	0.849	114 (70.4)	10 (55.6)	0.198	126 (69.2)
Redness	38 (25.7)	6 (17.6)	0.324	40 (24.7)	2 (11.1)	0.251	44 (24.2)
Swelling	41 (27.7)	5 (14.7)	0.116	43 (26.5)	1 (5.6)	0.078	46 (25.3)
Administration of acetaminophen	126 (85.1)	23 (67.6)	0.017 *	136 (84.0)	11 (61.1)	0.018 *	149 (81.9)
Duration of acetaminophen consumption, days (mean ± SD)	2.05 ± 2.54	1.21 ± 1.04	0.007 *	2.00 ± 2.45	1.00 ± 1.03	0.011 *	1.89 ± 2.35
Scores of AE severity (mean ± SD)							
Systemic SUM	10.8 ± 7.2	6.7 ± 7.4	0.004 *	10.6 ± 7.2	5.3 ± 7.4	0.006 *	10.1 ± 7.4
Systemic SoM	25.6 ± 26.6	13.2 ± 15.7	0.010 *	24.7 ± 25.8	9.4 ± 12.5	0.013 *	23.3 ± 25.4
Localized SUM	3.2 ± 3.4	2.4 ± 4.5	0.199	3.2 ± 3.3	1.6 ± 1.9	0.054	3.1 ± 3.2
Localized SoM	15.2 ± 24.0	8.5 ± 10.7	0.122	14.7 ± 23.0	4.3 ± 5.9	0.067	14.0 ± 22.2

* *p* < 0.05. Abbreviations: ELISA, enzyme-linked immunosorbent assay; SD, standard deviation; AE, adverse event; SUM, sum of symptoms’ severity score; SoM, sum of multiplying each symptoms’ severity by the duration of symptoms.

**Table 2 jcm-10-02844-t002:** The association between demographics and vaccine adverse events.

Demographics	Systemic SUM	Systemic SoM	Localized SUM	Localized SoM
Sex, mean ± SD				
*p*	0.001 *	<0.001 *	0.004 *	0.006 *
Male	7.6 ± 7.0	15.4 ± 17.2	2.2 ± 2.5	8.7 ± 14.3
Female	11.4 ± 7.2	27.6 ± 28.0	3.6 ± 3.5	16.9 ± 25.2
Occupation, mean ± SD				
*p*	0.159	0.028 *	0.763	0.626
Doctor	9.6 ± 7.0	20.5 ± 19.0	2.9 ± 3.1	13.3 ± 21.2
Nurse	11.7 ± 7.5	31.0 ± 32.3 ^a^	3.5 ± 3.6	17.0 ± 26.3
Laboratory technologist	8.1 ± 8.0	13.5 ± 17.6 ^a^	2.7 ± 2.5	8.7 ± 10.5
Researcher	4.0 ± 5.7	8.0 ± 11.3	3.5 ± 5.0	17.5 ± 24.7
Radiologist	6.0 ± 5.7	11.0 ± 12.7	0.5 ± 0.7	2.0 ± 2.8
Hospital administrative assistant	5.5 ± 7.8	11.0 ± 15.6	3.0 ± 2.8	5.5 ± 6.4

* *p* < 0.05. Abbreviations: SUM, sum of symptoms’ severity score; SoM, sum of multiplying each symptoms’ severity by the duration of symptoms; SD, standard deviation. ^a^ Significant difference was shown between nurses and laboratory technologists.

**Table 3 jcm-10-02844-t003:** Strength of the agreement of Euroimmun IgG ELISA and surrogate virus neutralization test.

		Euroimmun IgG ELISA					
Diagnostic Methods		Categorization of Borderline into Negative			Categorization of Borderline into Positive		
		Positive	Negative	κ	Positive	Negative	κ
sVNT	Positive	145	1	0.602	146	0	0.403
	Negative	17	17		24	10	

Abbreviations: ELISA, enzyme-linked immunosorbent assay; sVNT: Surrogate Virus Neutralization Test.

## Data Availability

Data is contained within the article.

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
