# Peer review of "Systemic Adverse Events and Use of Antipyretics Predict the Neutralizing Antibody Positivity Early after the First Dose of ChAdOx1 Coronavirus Disease Vaccine"

_jcm, 2021, doi:10.3390/jcm10132844_

Round 1
Reviewer 1 Report
The term Antipyretics was too generic. It would be interesting to report what kind of antipyretics were used paracetamol, a weak cyclooxigenase inhibitor or pyrazolones, strong cyclooxigenase inhibitors and their dosage. To administer paracetamol 500 mg is different than giving 1000 mg. Authors did not distinguish it during their study and it is too late recruiting back the patients
Author Response
- The term Antipyretics was too generic. It would be interesting to report what kind of antipyretics were used paracetamol, a weak cyclooxigenase inhibitor or pyrazolones, strong cyclooxigenase inhibitors and their dosage. To administer paracetamol 500 mg is different than giving 1000mg. Authors did not distinguish it during their study and it is too late recruiting back the patients.
- We recommended the 1-2 tablets of acetaminophen to HCWs if they would undergo the vaccine’s adverse event before vaccination. We collected the duration of acetaminophen (days), but not the type and dosage of the acetaminophen. Additional statistical analysis about the duration of antipyretics was conducted, and the results were added to Table 1.

Reviewer 2 Report
- How did authors ensure that past exposure to COVID-19 was not there as many individuals might be asymptomatic? Did they check pre-vaccination antibody tests similar to what they did after the vaccine? I think this point is important as we know that overall serological response is higher if a patient had past history of COVID-19 and if he/she also received COVID-19 vaccine subsequently.
- Did authors collect any data for any vaccine breakthrough infections which might have been asymptomatic, and could have let to a higher neutralizing antibody response?
Author Response
1. How did authors ensure that past exposure to COVID-19 was not there as many individuals might be asymptomatic? Did they check pre-vaccination antibody tests similar to what they did after the vaccine? I think this point is important as we know that overall serological response is higher if a patient had past history of COVID-19 and if he/she also received COVID-19 vaccine subsequently.
--> Under the Seoul Metropolitan Government’s executive order, medical staffs at hospital in Seoul have been required to conduct SARS-CoV-2 PCR tests to prevent hospital acquired infection every 2 weeks since March, 2021. Therefore, all HCWs conducted SARS-CoV-2 PCR test and all of them were confirmed negative around the time of the vaccination.
There could have been cases diagnosed with asymptomatic infections with COVID-19 in 2020. However, in Korea, cumulative incidence of COVID-19 is extremely low due to the legally strict social distancing and wearing mask. In addition, 2 of 5,284 (0.04%) people were undiagnosed people from the 21st of April to the 12th of December in 2020 and 4 of 2,248 (0.18%) people were underdiagnosed people from 20th of January to 30th of April in 2021 according to the nationwide health survey. Thus, the risk of enrolled undiagnosed people was very low.
Furthermore, serological survey to HCWs working at our hospital was done 6-7 months after the first case of COVID-19 in Korea, but no one was confirmed positive. We added the figure below according to the timeline of hygene policy in our hospital and box which is the period of serological survey with cumulative admitted COVID-19 patients in our hospital and weekly incidence of COVID-19 in Korea, Seoul, and our hospital.
2. Did authors collect any data for any vaccine breakthrough infections which might have been asymptomatic, and could have let to a higher neutralizing antibody response?
--> Under the Seoul Metropolitan Government’s executive order, medical staffs at hospital in Seoul have been required to conduct SARS-CoV-2 PCR tests to prevent hospital acquired infection every 2 weeks since March, 2021. Therefore, all HCWs conducted SARS-CoV-2 PCR test and all of them were confirmed negative around the time of the vaccination. No positive cases were confirmed in the every-2-week PCR tests so far.
We also added the description about the asymptomatic infection before vaccination and breakthrough infection in the section of discussion.
(Page 6-7, line 175-182) First, among HCWs enrolled in this study, some may have had COVID-19 that remained undetected during symptom-based testing, which may have affected antibody test results. However, in Korea, only 0.04% and 0.18% were diagnosed with COVID-19 even in large-scale sero-surveillance, perhaps due to the small scale of the COVID-19 outbreak in Korea due to thorough infection control and prevention. Moreover, from March 21, all HCWs in Seoul were instructed to undergo PCR tests every 2 weeks. Therefore, it is con-sidered that the probability of COVID-19 breakthrough infection among HCWs is very low.

Reviewer 3 Report
The article submitted by Ji Young Park and co-authors is a study regarding the adverse events and use of antipyretics after the first dose of ChAdOx1 COVID-19 vaccine.
The topic is of interest due to global spread that led to an increased number of infections and deaths.
However, in my opinion, the paper should better clarify how the severity of symptoms was graded? What are the symptoms that have fallen into the category mild, moderate, seveare ? I also recommend a clearear explanation of SUM and SoM, maybe a reference related to SUM and SoM.
I also suggest explaining why in the results only 180 and not 182 samples were analyzed by ELISA and clarify the differences in the results regarding the two type of testing: sVNT and ELISA (M: 48 vs.55, and F:100 vs. 107).
I also recommend providing possible explanation regarding the higher incidence/adverse events in women than in men.
I also suggest specifying the type of study, inclusion criteria, exclusion criteria.
Author Response
1. The paper should better clarify how the severity of symptoms was graded?
--> The severity of AEs in each individual was reported subjectively and the participants volunteered to be included in the study, which might have led to bias. We added the description about severity scoring.
(Page 2, line 56-7) Severity was subjectively graded from 1 to 5 according to the Faces Pain Scale.
2. What are the symptoms that have fallen into the category mild, moderate, severe?
--> It was not classified as ‘mild, moderate, or severe’ depending on the type of symptoms. We just scored on the severity of the symptoms.
3. I also recommend a clear explanation of SUM and SoM, maybe a reference related to SUM and SoM.
--> Thank you for the comment. There was no literature referred to the scoring AEs. However, the degree of AEs were evaluated by devising two methods; SUM and SoM, because it is important severity as well as duration of AEs. We also corrected the explanation of SUM and SoM.
(Page 2, line 47-49) The severity of symptoms was scored in two ways: the sum of each symptoms’ severity score (SUM) and the sum of multiplying each symptoms’ severity by the duration (days) (SoM=SUM [symptom’s severity × days]).
4. I also suggest explaining why in the results only 180 and not 182 samples were analyzed by ELISA and clarify the differences in the results regarding the two type of testing: sVNT and ELISA (M: 48 vs.55, and F:100 vs. 107).
--> Because of the lack of samples, 180 samples could be tested with ELISA.
5. I also recommend providing possible explanation regarding the higher incidence/adverse events in women than in men.
--> Thank you for the comment. We explained this phenomenon in the manuscript.
(Page 6, line 172-4) We also found that systemic and localized AE scores (SUM and SoM) were higher among females than among males. This phenomenon is attributed to unknown immuno-logical differences between the two sexes [1, 11].
6. I also suggest specifying the type of study, inclusion criteria, exclusion criteria.
--> We added the description about the type of study, inclusion criteria, and exclusion criteria. Thank you for your comment.
(Page 2, line 49-52) This study was conducted as a cross-sectional study. Among the 1356 HCWs who received the first dose of the ChAdOx1 vaccine at the Chung-Ang University Hospital, we enrolled HCWs who participated voluntarily in this study and excluded those who received BNT162b2 vaccine as well as those diagnosed with COVID-19.

Round 2
Reviewer 3 Report
The authors have made all the suggested modifications.